# A Causal Bandit Approach to Learning Good Atomic Interventions in Presence of Unobserved Confounders

**Aurghya Maiti**[1]     **Vineet Nair**[2]     **Gaurav Sinha**[1]

[1]Adobe Research, Bangalore, India
[2]Technion Israel Institute of Technology, Haifa, Israel

## Abstract

We study the problem of determining the best atomic intervention in a Causal Bayesian Network (CBN) specified only by its causal graph. We model this as a stochastic multi-armed bandit (MAB) problem with side-information, where interventions on CBN correspond to arms of the bandit instance. First, we propose a simple regret minimization algorithm that takes as input a causal graph with observable and unobservable nodes and in $T$ exploration rounds achieves $\tilde{O}(\sqrt{m(\mathcal{C})/T})$ expected simple regret. Here $m(\mathcal{C})$ is a parameter dependent on the input CBN $\mathcal{C}$ and could be much smaller than the number of arms. We also show that this is almost optimal for CBNs whose causal graphs have an $n$-ary tree structure. Next, we propose a cumulative regret minimization algorithm that takes as input a causal graph with observable nodes and performs better than the optimal MAB algorithms that do not use causal side-information. We experimentally compare both our algorithms with the best known algorithms in the literature.

## 1 INTRODUCTION

Causal Bayesian Networks or CBNs [Pearl, 2000] have become the natural choice for modelling causal relationships in many real-world situations such as price-discovery [Haigh and Bessler, 2004], computational-advertising [Bottou et al., 2013], healthcare [Velikova et al., 2014], etc. A CBN has two components: a directed acyclic graph (DAG) called the causal graph, and conditional probability distributions of each node given its parents such that the joint distribution of all variables factorizes as a product of these conditionals. Edges in the causal graph represent direct causal relationships and therefore it captures the data generation process.

In its most general setup, only a subset of the variables appearing in the CBN are observable and the rest are unobserved (see Definition 1.3.1 in Pearl [2000]). CBNs enable modelers to simulate the effect of external manipulations via a process called *intervention*. An intervention forcibly fixes selected observable variables in the graph and breaks the edges coming into them. Data generated from the resulting model is the simulated outcome of the intervention. In the presence of an outcome variable of interest $Y$ (assumed to be observable), a natural question (see epidemic prevention example below) is to find the variable $X$ and a corresponding value $x$, such that the intervention setting $X$ to $x$ leads to the maximum expected value of $Y$ i.e. $X = x$ has the highest causal impact on $Y$. Such an intervention which manipulates only a single variable is called an *atomic intervention*.

The problem of learning the best atomic intervention was formulated as a sequential decision making problem called *Causal Bandits* (CB) in Lattimore et al. [2016]. In CB, access to the underlying DAG of the CBN is assumed but the associated conditional probability distributions are unknown. The outcome variable $Y$ is considered as a reward variable and the set of allowed atomic interventions are modelled as arms of a bandit instance. In addition, there is an *observational arm* corresponding to the empty intervention, and pulling the observational arm generates a sample from the joint distribution of all observable variables. Here, identifying the best atomic intervention is equivalent to the well-studied *best-arm identification* problem in a multi-armed bandit (MAB) instance. However, in CB, an algorithm while pulling an arm has access to *causal* side information derived from the causal graph associated with the input CBN. See Lattimore et al. [2016] and the references therein for a comparison of CB and MAB problems with other types of side-information.

In this work, we study CB for causal graphs with unobserved confounders. (UCs). These are unobserved variables that are parents of at least two observable variables. To the best of our knowledge, this is the first work that analyses the regret of causal bandit algorithms when input causal graphs contain UCs. Moreover, in the fully observable setting, i.e.

*Accepted for the 38th Conference on Uncertainty in Artificial Intelligence* (UAI 2022).

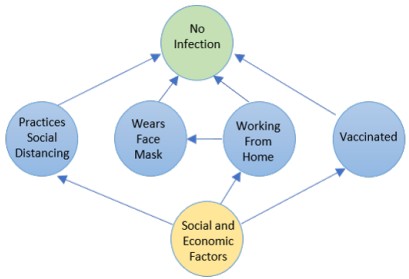

Figure 1: Causal Graph: Epidemic Prevention with Social and Economic Factors being an Unobserved Confounder

when all variables are observable, our algorithm does not assume any structural constraints on the input causal graph and hence can be applied quite generally. Before stating our contributions, we provide a motivating example where determining the best atomic intervention is important. Suppose a policy-maker is required to identify the best precautionary measure that should be enforced to reduce spread of a disease. The available measures are mandating social distancing, wearing of face mask, making people work from home and preventive vaccinations. Since the effect of each measure needs to be isolated while disrupting public life minimally, the policy-maker can enforce at most one of these measures at a given time. The policy-maker can conduct surveys to collect data from public about which measures were taken by them (other than the one enforced) and whether they got infected or not. The goal would then be to design a mechanism of implementing such enforcement one by one, during a time period and collecting the respective survey data to identify the best measure to enforce. Note that, using domain knowledge of health experts, policy makers can have access to an underlying causal graph such as the one in Fig. 1. They would want to use this graph to decide if and when a particular measure should be enforced.

## 1.1 OUR CONTRIBUTIONS

We study CB with respect to two standard objectives in MAB: simple and cumulative regret. Simple regret captures the best arm identification problem described above, whereas the cumulative regret is more natural when the goal is to maximize the cumulative reward at the end of $T$ rounds instead of determining the best arm. We state our contributions below; meanings of the relevant terminologies are defined in Sec. 2.

**Simple Regret Minimization**: We propose a simple regret minimization algorithm called SRM-ALG. The input causal graph of the underlying CBN $\mathcal{C}$ is assumed to be (without loss of generality) a Semi-Markovian Causal Graph or SMCG (defined in Sec. 2) on observable nodes having both directed and bi-directed edges (representing presence of UCs) between the nodes. We assume that the input SMCG

is identifiable with respect to a set of intervenable nodes $\mathbf{X}$, meaning that the interventional distributions arising from atomic interventions on variables in $\mathbf{X}$ can be consistently estimated from the observational data itself (see definition in Sec. 2). When the c-components (connected components of bi-directed edges, defined in Sec. 2) of the SMCG are bounded in size (by a constant) and the total in-degree of all vertices in the c-components are also bounded, given a time budget of $T$ rounds, SRM-ALG attains $\tilde{O}(\sqrt{m(\mathcal{C})/T})$ expected simple regret (see Thm. 3.1). Here, $m(\mathcal{C})$ depends on the input CBN and is $\leq 2N$, where $N$ is the number of intervenable nodes i.e. $N = |\mathbf{X}|$.

In Sec. 3 we give examples of graphs, where $m(\mathcal{C}) \ll N$, and hence SRM-ALG performs better than standard bandit algorithms which achieve $\Omega(\sqrt{N/T})$ expected simple regret in the worst-case (Thm. 4 in Audibert et al. [2010]). SRM-ALG leverages the causal side-information available by deriving reward estimates for each arm from pulls of the observational arm. The quality of these derived estimates depends on the input CBN. The quantity $m(\mathcal{C})$ intuitively captures the trade-off between the number of arms with bad estimates, and the quality of estimates determined from intervening upon them explicitly.

We note that Lattimore et al. [2016] and Nair et al. [2021] propose algorithms in the fully observable setting, when the input causal graph is a parallel graph and a no-backdoor graph, respectively.[1] For these special classes of graphs, SRM-ALG recovers the regret guarantees given in Lattimore et al. [2016], Nair et al. [2021]. Hence, SRM-ALG can be viewed as a *significant* generalization of these algorithms to more general causal graphs with UCs. Further, Yabe et al. [2018] proposed a causal bandit algorithm for interventions which can simultaneously manipulate multiple variables. However, the input causal graph is assumed to have no UCs, and regret guarantee of their algorithm is $\tilde{O}(\sqrt{N/T})$ for atomic interventions. In particular, its performance is not better than optimal MAB algorithms that do not take causal side-information into account. In Sec. 6, we experimentally compare the regret guarantee of SRM-ALG with the algorithm in Yabe et al. [2018], as well as MAB algorithms that do not take causal side-information into account.

**Lower Bound on Simple Regret**: We also show that SRM-ALG is almost optimal for CBNs associated with a large and important class of causal graphs. Specifically, in Thm. 4.1, we show that for any causal graph $\mathcal{G}$ which is an $n$-ary tree on $N + 1$ nodes[2], and any $M \in [1, N]$ there is a probability distribution $\mathbb{P}$ compatible with the the causal graph such that $m(\mathcal{C}) = M$, and the expected simple regret of any algorithm at the end of $T$ rounds is $\Omega(\sqrt{M/T})$[3]. We remark that these graphs naturally capture important CBNs

---

[1] These graphs have no backdoor paths from any $X \in \mathbf{X}$ to $Y$, implying $\mathbb{P}(Y \mid do(x)) = \mathbb{P}(Y \mid x)$.

[2] $N$ nodes are intervenable and can causally effect node $Y$.

[3] Here $\mathcal{C}$ is the CBN $(\mathcal{G}, \mathbb{P})$ and $m(\mathcal{C})$ is as described before.

like causal trees [Greenewald et al., 2019]. Also, the class of graphs considered in Thm. 4.1 subsumes the parallel graph model, and for them lower bound in Thm. 4.1 matches the lower bound given in Lattimore et al. [2016]. Importantly, Thm. 4.1 implies that the regret guarantee of `SRM-ALG` can be only improved by considering more nuanced structural restrictions on the causal graph, which could enable more causal information sharing between the interventions.

**Cumulative Regret Minimization**: We propose a cumulative regret minimization algorithm called `CRM-ALG`. All variables in the input causal graph are assumed to be observable. `CRM-ALG` achieves constant expected cumulative regret if the observational arm is optimal, and otherwise achieves better regret than the optimal `MAB` algorithm which does not take causal side-information into account (see Thm. 5.1). Cumulative regret minimization in general graphs were also studied by Lu et al. [2020] and Nair et al. [2021]. However, they crucially assume that distribution of parents of the reward node is known for every intervention. `CRM-ALG` does not make this assumption. The reason why we develop `CRM-ALG` in the fully observable setting (unlike `SRM-ALG`) is rather technical and is explained at the end of Sec. 5.

## 1.2 RELATED WORK

As noted before, causal bandits was introduced in Lattimore et al. [2016], where an almost optimal algorithm was proposed for CBNs associated with a parallel causal graph. Recently, a similar algorithm for simple regret minimization along with an algorithm for cumulative regret minimization was proposed for no-backdoor graphs in Nair et al. [2021], and the observation-intervention trade-off was studied when interventions are costlier than observations. An importance sampling based algorithm was proposed by Sen et al. [2017a] to minimize simple regret but only soft-interventions at a single node were considered. The cumulative regret minimization problem for general causal graphs was studied in Lu et al. [2020], Nair et al. [2021], but they assume the knowledge of the distributions of the the parents of the reward variable for every intervention. Recently Lu et al. [2021a] designed a cumulative regret minimization algorithm which only utilizes the side information that the underlying causal graph is a directed tree or a causal forest (and does not require the exact DAG). Assuming faithfulness and identifiability, their algorithm outperforms the standard MAB algorithms. Sen et al. [2017b] studied the contextual bandit problem where the observed context influences the reward via a latent confounder variable, and proposed an algorithm with better guarantee compared to standard contextual bandit. Lee and Bareinboim [2018, 2019] gave a procedure to compute the minimum possible intervention set by removing sub-optimal interventions identifiable from the input causal graph, and they empirically demonstrated that ignoring such information leads to huge regret. Recently,

Lu et al. [2021b] introduced the causal Markov decision processes, where at each state a causal graph determines the action set, and gave algorithms that achieve better policy regret when the causal side-information is taken into account. Finally, in a related line of work Bareinboim et al. [2015] promote the use of observational data for bandit problems in the presence of UCs. We note that our proposed algorithms `SRM-ALG` and `CRM-ALG` both use observational samples to leverage side-information and hence achieve better regrets.

## 2 NOTATIONS AND PRELIMINARIES

**Notations:** For positive integers $m, n$ with $m < n$, $[n]$ denotes the set $\{1, \ldots, n\}$ and $[m, n]$ denotes the set $\{m, m+1, \ldots, n\}$. Random variables are denoted using capital letters (e.g. $X$) and corresponding lower case letters (i.e. $x$) denote the assignment $X = x$. Unless otherwise specified, all random variables will be discrete with finite support. Sets of random variables are denoted by bold face letters (e.g. $\mathbf{X}$) and corresponding bold face lower case letter (i.e. $\mathbf{x}$) denotes the assignment $\mathbf{X} = \mathbf{x}$. We use $\mathbb{P}(\mathbf{X} = \mathbf{x})$ (equivalently $\mathbb{P}(\mathbf{x})$) to denote the probability of $\mathbf{X}$ taking the value $\mathbf{x}$. Conditional probability of $\mathbf{X} = \mathbf{x}$ given $\mathbf{Y} = \mathbf{y}$ is denoted by $\mathbb{P}(\mathbf{x} \mid \mathbf{y})$. Size of any set $S$ is denoted by $|S|$.

**Causal Bayesian Network:** A Bayesian Network or BN is a tuple $(\mathcal{G}, \mathbb{P})$, where $\mathcal{G} = (\mathbf{V}, \mathbf{E})$ is a directed acyclic graph (DAG), and $\mathbf{V} = \{V_1, \ldots, V_n\}$ and $\mathbf{E}$ are the set of nodes and edges in $\mathcal{G}$ respectively. A node $V_i$ is called the parent of $V_j$ and $V_j$ the child of $V_i$, if there is a directed edge from $V_i$ to $V_j$ in $\mathbf{E}$. The nodes in $\mathbf{V}$ are labelled by random variables, and $\mathbb{P}$ is the joint distribution of $\mathbf{V}$ that factorizes over $\mathcal{G}$, i.e. $\mathbb{P}(\mathbf{V}) = \prod_{i=1}^{n} \mathbb{P}(V_i \mid \mathbf{Pa}(V_i))$, where $\mathbf{Pa}(V_i)$ is the set of parents of $V_i$. Sometimes, in a BN, certain nodes are not observable and are termed unobserved variables. In this situation, for each node $V_i \in \mathbf{V}$, $\mathbf{Pa}(V_i)$ will denote the set of *observable* parents of $V_i$. A **Causal Bayesian Network** or CBN is a BN where each edge denotes an immediate causal relationship. The graph $\mathcal{G}$ (called the **Causal Graph**) corresponding to a CBN describes the data generation process not just of the observational distribution $\mathbb{P}$ but also of interventional distributions that can be derived from it. An intervention on an observable node $X \in \mathbf{V}$ is denoted as $do(X = x)$, where $X$ is set to value $x$ and all the edges coming in to $X$ are removed. The resulting graph defines a probability distribution $\mathbb{P}(\mathbf{V} \setminus \{X\} \mid do(X = x))$ over $\mathbf{V} \setminus \{X\}$, called an interventional distribution. In the presence of unobserved variables it is convenient to assume (without loss of generality) [Tian and Pearl, 2002b, Verma and Pearl, 1988] that the underlying graph $\mathcal{G}$ of the CBN is semi-markovian. Formally, a **Semi-Markovian Causal Graph** or an SMCG is a DAG, where every unobserved variable is a root and has exactly two observable children [Tian and Pearl, 2002b, Acharya et al., 2018]. These unobserved variables are called **Unobserved Confounders** or UCs in

the rest of the paper. It is convenient to represent SMCGs with observable vertices only by adding a bi-directed edge between two observable vertices if they have a common unobserved parent and removing the unobserved parent from the graph [Tian and Pearl, 2002b]. Such graphs thus comprise of both directed and bi-directed edges with all vertices $\mathbf{V} = \{V_1, \ldots, V_n\}$[4] observable. The bi-directed edges can be used to partition the observable vertices into what are called **c-components**[5] [Tian and Pearl, 2002b]. Two observable vertices are said to be in the same c-component if and only if they are connected by a path of bi-directed edges. Let $\mathbf{X} = \{X_1, \ldots, X_N\} \subset \mathbf{V}$ denote the set of intervenable nodes. The c-component containing the node $X_i$ is denoted by $S_i$ and its size is denoted by $k_i$ i.e. $k_i = |S_i|$. The number of observable parents of $X_i$ is denoted by $d_i$, i.e. $d_i = |\mathbf{Pa}(X_i)|$. We define $\mathbf{Pa}^+(S_i) = S_i \cup \bigcup_{V \in S_i} \mathbf{Pa}(V)$, and $\mathbf{Pa}^c(X_i) = \mathbf{Pa}^+(S_i) \setminus \{X_i\}$. For more details on these definitions we refer the reader to Tian and Pearl [2002b], Verma and Pearl [1988], Acharya et al. [2018]. An important question that arises in the context of SMCGs is that of **identifiability**, which asks whether the interventional distributions $\mathbb{P}(\mathbf{V} \setminus \{X\} \mid do(X = x))$ can be estimated consistently using observational data sampled from $\mathbb{P}(\mathbf{V})$. In the case of atomic interventions, Tian and Pearl [2002a] provided a necessary and sufficient condition for this to happen. They show that (Thm. 3 in Tian and Pearl [2002a]) $\mathbb{P}(\mathbf{V} \setminus \{X\} \mid do(X = x))$ is identifiable if and only if there is no bi-directed path connecting $X$ to any of its children. For this work, we say that an "SMCG is identifiable with respect to variables in $\mathbf{X}$" if the identifiability condition mentioned above holds for all intervenable nodes in $\mathbf{X}$. Note that, when all variables are observable, there are no bi-directed paths and the interventional probabilities are always identifiable. Moreover, in the observable setting one can use the *backdoor criterion* [Pearl, 2009] for estimating the interventional probabilities from observational samples. In the general setting, Bhattacharyya et al. [2020] provides an efficient procedure (based on construction in Tian and Pearl [2002a]) to estimate the interventional distribution using observational samples which we use in Sec. 3.

**Causal Bandits:** A causal bandit algorithm receives as input a causal graph $\mathcal{G} = (\mathbf{V}, \mathbf{E})$ (corresponding to some CBN $\mathcal{C}$), the associated set of (binary) intervenable nodes $\mathbf{X} \subseteq \mathbf{V}$ and the designated (observable) reward node $Y \in \mathbf{V}$. We assume there are $N$ intervenable nodes $\mathbf{X} = \{X_1, \ldots, X_N\}$, and there are $2N$ interventions denoted $a_{i,x} = do(X_i = x)$ for $i \in [N]$ and $x \in \{0, 1\}$. The empty intervention $do()$, which corresponds to the observational distribution is denoted as $a_0$. These $2N + 1$ interventions constitute the arms $\mathcal{A} = \{a_{i,x} \mid i \in [N], x \in \{0,1\}\} \cup \{a_0\}$

---

[4]By abuse of notation we denote the set of observable vertices by $\mathbf{V}$ from here on wards.

[5]If a node is not incident by any bi-directed edge then its c-component is itself.

of the bandit instance. A causal bandit algorithm is a sequential decision making process that at each time $t$, makes an intervention $a_t \in \mathcal{A}$, and observes the sampled values of the nodes in $\mathbf{V}$ including the value of the node $Y$. The values of nodes $V \in \mathbf{V}$, $X \in \mathbf{X}$ and $Y$ sampled at time $t$ are denoted as $V(t), X(t)$, and $Y(t)$ respectively. Throughout the paper we use $i, x$, and $a$ to index the sets $[N], \{0, 1\}$, and $\mathcal{A}$ respectively. The expected reward corresponding to intervention $a_{i,x} \in \mathcal{A}$ and $a_0 \in \mathcal{A}$ is denoted as $\mu_{i,x} = \mathbb{E}[Y \mid do(X_i = x)]$ and $\mu_0 = E[Y]$. We study the causal bandit problem with respect two standard objectives in bandit literature: simple and cumulative regret.

**Simple Regret**: The expected simple regret of an algorithm ALG that outputs arm $a_T$ at the end of $T$ rounds is defined as $r_{\text{ALG}}(T) = \max_{a \in \mathcal{A}} \mu_a - \mu_{a_T}$.

**Cumulative Regret**: Let ALG be an algorithm that plays arm $a_t$ at time $t \in [T]$. Then the expected cumulative regret of ALG at the end of $T$ rounds is defined as $R_{\text{ALG}}(T) = \max_{a \in \mathcal{A}} \mu_a \cdot T - \sum_{t \in [T]} \mu_{a_t}$.

Throughout this paper, we assume that the intervenable nodes are binary, distribution of any intervenable node $X_i$ conditioned on its parents $\mathbf{Pa}(X_i)$ is Bernoulli. We assume without loss of generality that $X_i \prec Y$ where $\prec$ is a topological order on $\mathcal{G}$. In SRM-ALG (Sec. 3) we assume that the input is an SMCG that is identifiable with respect to the intervenable variables $\mathbf{X}$ and in CRM-ALG (Sec. 5), we assume the input causal graph has no UCs and the underlying distribution $\mathbb{P}$ is strictly positive i.e. $\mathbb{P}(\mathbf{v}) > 0$ for all $\mathbf{v}$.

Other than the above, our algorithms do not make any other structural assumptions and are therefore significantly general compared to the previous works [Lattimore et al., 2016, Lu et al., 2020, Nair et al., 2021, Lu et al., 2021a]. However, we would like to note that the results in our main theorems i.e. Thms. 3.1, 4.1, 4.2, 5.1 are stated assuming that all the c-components have bounded size implying that $k_i = O(1)$ for all $i \in [N]$ and that the total number of observable parents of nodes in c-component $S_i$, i.e. $|\cup_{V \in S_i} \mathbf{Pa}(V)|$ is also bounded above by a constant. Note that this clearly implies that the indegree $d_i = |\mathbf{Pa}(X_i)|$ is also $O(1)$. These help us describe the results more cleanly highlighting the main parameters of importance for this work. But our algorithms work even without these assumptions. We note that these assumptions are common in the causal inference literature [Acharya et al., 2018, Bhattacharyya et al., 2020].

# 3 SIMPLE REGRET MINIMIZATION

In this section, we state and analyze our simple regret minimization algorithm called SRM-ALG which takes as input an SMCG which is identifiable with respect to intervenable variables $\mathbf{X}$ (See Sec. 2 for definition). Our proposed algorithm repeatedly plays the observational arm $a_0$ for the first $T/2$ rounds. Using this observational data, it determines

a small set of arms to pull (i.e. perform interventions) in the next $T/2$ rounds and estimates their rewards using the interventional samples thus obtained. Finally, for the arms it does not pull, it uses the collected observational samples (from the initial $T/2$ pulls of $a_0$) to estimate their rewards by adapting a procedure from Bhattacharyya et al. [2020] which efficiently estimates distributions resulting from an atomic intervention using observational samples. We remark that previous works in Lattimore et al. [2016] and Nair et al. [2021] imposed structural restrictions on the input causal graphs which allowed observational samples to be directly used for estimating rewards corresponding to interventions[6]. SRM-ALG, on the other hand, can work with more general identifiable SMCGs and still estimate rewards of multiple arms simultaneously using the observational arm pulls. SRM-ALG is presented in Algorithm 1. We explain each step below.

---

**Algorithm 1** SRM-ALG

---

INPUT: Causal Graph $\mathcal{G} = (\mathbf{V}, \mathbf{E})$, set of intervenable nodes $\mathbf{X} \subseteq \mathbf{V}$ and time horizon $T$.

1: His = {} /* His would be used to keep the history of sampled values in the first $T/2$ rounds. */
2: **for** $t \in [1, \ldots, T/2]$ **do**
3:     Play arm $a_0$ and let His = His $\cup$ {$\mathbf{V(t)} \setminus \mathbf{U}(t), Y(t)$}.
4: For each $i \in [n]$, compute $\widehat{q}_i$ (as defined in Equation 2) and $\widehat{m}$ (as an estimate of $m$) by plugging in $\widehat{q}_i$ in place of $q_i$ in Equation 1. Let $\mathcal{Q} = \{a_{i,x} \in \mathcal{A} : \widehat{q}_i^{k_i} < 1/\widehat{m}\}$.
5: **for** $a_{i,x} \in \mathcal{Q}$ **do**
6:     Play arm $a_{i,x}$ and observe $Y$ for $\frac{T}{2|\mathcal{Q}|}$ rounds.
7:     Estimate reward as $\widehat{\mu}_{i,x} = \frac{2|\mathcal{Q}|}{T} \sum_{t=1}^{T/2|\mathcal{Q}|} Y(t)$.
8: **for** $a_{i,x} \notin \mathcal{Q}$ **do**
9:     For each $i \in [n], x \in \{0, 1\}$, use Algorithm C.1 in App. C with inputs $\mathcal{G}$, His to get reward estimate $\widehat{\mu}_{i,x}$.
10: Return estimated optimal $a_T^* \in \text{arg-max}_{a \in \mathcal{A}} \widehat{\mu}_a$.

---

*Steps 1–4*: At Steps $1 - 3$, SRM-ALG collects $T/2$ observational samples from pulls of $a_0$ and at Step 4 it identifies a set of arms $\mathcal{Q}$ whose reward estimates (when computed using the collected observational samples) will be bad[7]. This is done using a quantity $m(\mathcal{C})$ defined next; the meaning of relevant notations can be found in Sec. 2. Let $q_i = \min_{\mathbf{z},x} \mathbb{P}(X_i = x, \mathbf{Pa}^c(X_i) = \mathbf{z})$. For each $\tau \in [2, 2N]$, let $I_\tau = \{i : q_i^{k_i} < 1/\tau\}$[8]. We define,

$$m(\mathcal{C}) = \min\{\tau : |I_\tau| \leq \tau\}. \tag{1}$$

The observational samples that were collected are used to

---

[6]The restrictions ensured that the conditional distributions are equal to the corresponding interventional distributions.
[7]Can be seen easily using Lemma D.3.
[8]Recall from Sec. 2 that $k_i$ is size of the c-component of $X_i$.

first compute estimates $\widehat{q}_i$ of $q_i$ given as:

$$\widehat{q}_i = \left(\frac{2}{T}\right) \cdot \min_{\mathbf{z},x} \left\{ \sum_{t=1}^{T/2} \mathbb{1}\{X_i(t) = x, \mathbf{Pa}^c(X_i)(t) = \mathbf{z}\} \right\} \tag{2}$$

These estimates are then plugged into the above definition of $m(\mathcal{C})$ to obtain it's estimate $\widehat{m}$. Finally the set of arms $\mathcal{Q}$ is defined as $\mathcal{Q} = \{a_{i,x} \in \mathcal{A} : \widehat{q}_i^{k_i} < 1/\widehat{m}\}$.

*Steps 5–10*: Since, using observational samples reward estimates of arms in $\mathcal{Q}$ will be bad, in Steps $5 - 7$, we pull these arms equal number of times by performing the corresponding interventions and estimate their rewards directly from the interventional samples. The observational samples collected in first $T/2$ rounds are used to compute the estimates for each arm $a_{i,x} \notin \mathcal{Q}$ at Steps $8, 9$. Reward estimates of these arms are computed using Algorithm C.1, App. C. Finally in Step 10, we return arm $a_{i,x}$ with the best reward estimate. Even though Algorithm C.1 uses Bhattacharyya et al. [2020], which assumes strong positivity, we do not need to explicitly make this assumption since low probability arms $a_{i,x}$ get pulled (by intervention) in Step 6 and only high probability arms are estimated using Algorithm C.1.

**Some remarks about $m(\mathcal{C})$:** Our definition of $m(\mathcal{C})$ above is a novel extension of a similar quantity $m$ defined in Lattimore et al. [2016] and reduces to $\Theta(m)$ for parallel graphs [Lattimore et al., 2016] and no-backdoor graphs [Nair et al., 2021]. As a result, the regret guarantee of SRM-ALG for these special classes of graphs matches those of the respective algorithms in these works. Operationally, $m(\mathcal{C})$ determines for us the optimal number of arms to be pulled in Steps $5 - 7$, in order to minimize expected regret. In particular, $I_{m(\mathcal{C})}$ is a set of arms such that the best arm in it (found using $T/2$ rounds of interventions) and the best arm in its complement $I_{m(\mathcal{C})}^c$ (found using $T/2$ rounds of observations) have reward estimates of similar quality.

We show that the expected simple regret of SRM-ALG in Theorem 3.1 stated below is $\tilde{O}(\sqrt{m(\mathcal{C})/T})$, which is an instance-dependent regret guarantee as $m(\mathcal{C})$ depends on the input CBN. If $m(\mathcal{C}) \ll N$ then SRM-ALG performs better than the optimal MAB algorithm. In particular, SRM-ALG explores only at most $2\widehat{m}+1$ arms compared to the $2N$ arms that must be explored by a standard best-arm identification MAB algorithm which achieves $\Omega(\sqrt{N/T})$ expected worst-case simple regret [Audibert et al., 2010]. It is easy to see that there are CBNs $\mathcal{C}$ with $m(\mathcal{C}) \ll N$ as illustrated in App. B. The proof of Theorem 3.1 is given in App. D.

**Theorem 3.1**
*The expected simple regret of* SRM-ALG *at the end of $T$ rounds is* $r_{\text{SRM-ALG}}(T) = O\left(\sqrt{\frac{m(\mathcal{C})}{T} \log \frac{NT}{m(\mathcal{C})}}\right)$.

**Remark:** The constant involved in the regret expression

is exponential in $\max_i\{k_i\}$ and $\max_i\{|\mathbf{Pa}^c(X_i)|\}$. Recall that these are constant as per our assumptions in Sec. 2.

# 4 SIMPLE REGRET LOWER BOUND

A closer inspection of SRM-ALG given in Sec. 3 reveals that the algorithm only leverages causal side-information while pulling the observational arm. Hence, there remains a possibility that a better algorithm could be designed which uses the information shared between any two interventions. In this section, we show that this is not possible for a large and important class of causal graphs that we call tree-graphs and denote it as T. Each graph in T is an $n$-ary tree, where each node can have 2 to $n$ children. Additionally, all the leaves are connected to the outcome node $Y$. We also assume that all nodes of any graph in T are observable. Note that a causal bandit algorithm receives as input a causal graph $\mathcal{G}$ (corresponding to some CBN $\mathcal{C} = (\mathcal{G}, \mathbb{P})$) but the associated distribution $\mathbb{P}$ is unknown to the algorithm. Since there are multiple probability distributions that are compatible with a given $\mathcal{G}$ the algorithm is required to learn the unknown $\mathbb{P}$ through the arm pulls. We show in Thm. 4.1 that for any causal graph $\mathcal{G}$ in T and any positive integer $M \leq N$, there exists a distribution $\mathbb{P}$ such that $M = m(\mathcal{C})$, where $\mathcal{C}$ is CBN $(\mathcal{G}, \mathbb{P})$, and, any algorithm must explore at least $\Omega(M)$ arms to minimize the worst-case expected simple regret.

**Theorem 4.1**
*Corresponding to every causal graph $\mathcal{G} \in$ T, with $N$ intervenable nodes and any positive integer $M \leq N$, there exists a probability measure $\mathbb{P}$ and CBN $\mathcal{C} = (\mathcal{G}, \mathbb{P})$ such that $m(\mathcal{C}) = M$ and the expected simple regret of any causal bandit algorithm ALG is $r_{ALG}(T) = \Omega\big(\sqrt{m(\mathcal{C})/T}\big)$.*

The proof of Thm. 4.1 is in App. E. Recall, from Sec. 3 that $m(\mathcal{C})$ is completely defined by $\mathbf{q} = (q_1, \ldots, q_N)$ and $\mathcal{G}$; in particular the definition of $m(\mathcal{C})$ does not depend on the entire probability distribution corresponding to CBN $\mathcal{C}$. We conclude this section by showing in Thm. 4.2 that the dependence of the regret on $\mathbf{q}$ in the definition of $m(\mathcal{C})$ is optimal for certain graphs. In Thm. 4.2, a $\mathbf{q}$ is valid if there exists a probability measure $\mathbb{P}$ for the graph $\mathcal{G}$, which results in the given $\mathbf{q}$. The proof of Thm. 4.2 is in App. F.

**Theorem 4.2**
*There exists a fully observable causal graph $\mathcal{G}$ with $N \geq 3$ nodes such that given any valid $\mathbf{q}$ corresponding to $\mathcal{G}$, there is a probability measure $\mathbb{P}$ conforming with $\mathbf{q}$ and CBN $\mathcal{C} = (\mathcal{G}, \mathbb{P})$ for which expected simple regret of any causal bandit algorithm is $\Omega\big(\sqrt{m(\mathcal{C})/T}\big)$.*

# 5 CUMULATIVE REGRET MINIMIZATION

In this section, we propose CRM-ALG, an algorithm based on the well-known UCB algorithm [Auer et al., 2002], that

sequentially performs (atomic) interventions and minimizes the cumulative regret incurred over the time horizon $T$. Unlike SRM-ALG, here we assume all nodes in the input graph $\mathcal{G}$ are observable and the joint distribution $\mathbb{P}$ is strictly positive[9]. Similar to the UCB family of algorithms CRM-ALG maintains UCB estimates at each round and pulls the arm with the highest UCB estimate. CRM-ALG performs better than the standard UCB algorithm [Auer et al., 2002] by leveraging (via *backdoor criterion* [Pearl, 2009]) the available causal side-information. In particular, CRM-ALG uses the samples from the observational arm pulls in addition to the samples from the arm pulls of $a_{i,x}$ to compute UCB estimates of $a_{i,x}$. Note that even though the observational arm may not be reward optimal, pulling it gives a simultaneous causal side-information about all the arms. CRM-ALG ensures a good trade-off between such a simultaneous exploration and the possible loss in reward by ensuring that $a_0$ is pulled at least a pre-specified (carefully chosen) number of times. We note that CRM-NB-ALG proposed for no-backdoor graphs in Nair et al. [2021], also ensures that the observational arm $a_0$ is pulled a pre-specified number of times, but CRM-ALG differs from CRM-NB-ALG on how the UCB estimates for the arms are computed at the end of each round. Next, we present the details of CRM-ALG.

---

**Algorithm 2** CRM-ALG (Minimizing cumulative regret in general causal graph)

---
INPUT: Causal graph $\mathcal{G}$ and the set of intervenable nodes
1: Pull each arm once and set $t = 2N + 2$
2: Let $\beta = 1$
3: **for** $t = 2N + 2, 2N + 3, \ldots$ **do**
4:     **if** $N_{t-1}^0 < \beta^2 \log t$ **then**
5:         Pull $a_t = a_0$
6:     **else**
7:         Pull $a_t = \arg\max_{a \in A} \bar{\mu}_a(t-1)$
8:     $N_t^a = N_{t-1}^a + \mathbb{1}\{a_t = a\}$
9:     Update $\hat{\mu}_a(t)$ and $\bar{\mu}_a(t)$ for all $a \in A$ according to Equations 3, 5, 6 and 7.
10:     Let $\hat{\mu}^* = \max_a \hat{\mu}_a(t)$
11:     **if** $\hat{\mu}_0(t) < \hat{\mu}^*$ **then**
12:         Set $\beta = \min\{\frac{2\sqrt{2}}{\hat{\mu}* - \hat{\mu}_0(t)}, \sqrt{\log t}\}$
13:     $t = t + 1$

---

We use $N_t^{i,x}$ and $N_t^0$ to denote the number of times arms $a_{i,x}$ and $a_0$ have been played at the end of $t$ rounds respectively, and further let $a_t$ denote the arm pulled at round $t$. Also, $\hat{\mu}_{i,x}(t)$ and $\bar{\mu}_{i,x}(t)$ (respectively $\hat{\mu}_0(t)$ and $\bar{\mu}_0(t)$) denotes the empirical and UCB estimates of the arm $a_{i,x}$ (respectively arm $a_0$) computed at the end of round $t$. At Step 4 CRM-ALG checks if the observational arm is pulled at least $\beta^2 \log t$ times, and accordingly either plays the ob-

---

[9]Strict positivity of the joint distribution is often assumed in the causality literature [Hauser and Bühlmann, 2012].

servational arm or the arm with the highest UCB estimate. Here the value of $\beta$ is updated as in Steps 11-12 . As noted before, the chosen update for $\beta$ and the corresponding pre-specified number of pulls for arm $a_0$ delicately balances the exploration-exploitation trade-off in expectation. The empirical estimate for arm $a_0$ at Step 9 is computed as follows

$$\widehat{\mu}_0(t) = \frac{1}{N_t^0} \sum_{s=1}^{t} \mathbb{1}\{Y(s) = 1, a_s = a_0\} . \qquad (3)$$

The empirical estimate for arm $a_{i,x}$ is involved, and as mentioned before is done by leveraging the following backdoor criterion (see Thm. 3.3.2 in Pearl [2009]).

$$\mathbb{P}(Y = 1 \mid do(X_i = x)) = \qquad (4)$$
$$\sum_{\mathbf{z}} \mathbb{P}(Y = 1 \mid X_i = x, \mathbf{Pa}(X_i) = z)\mathbb{P}(\mathbf{Pa}(X_i) = \mathbf{z})$$

Let the set of time steps $s \leq t$ at which arm $a_0$ is pulled be denoted by $S_t = \{t_1, \dots, t_{N_t^0}\}$. Partition $S_t$ into two parts $O_t$ containing all the time steps with odd indices (i.e. $t_1, t_3$, etc.) and $E_t$ containing all the time steps with even indices (i.e. $t_2, t_4$, etc.). We will now define some sets and intermediate estimators in order to describe the final estimator. Since $X_i$ is clear from the context, we do not use $i$ to index these intermediate estimators. In general these sets and estimators will be different for different $i$. We use time steps in $O_t$ to estimate $\mathbb{P}(Y = 1 \mid X_i = x, \mathbf{Pa}(X_i) = \mathbf{z})$, and those in $E_t$ to estimate $\mathbb{P}(\mathbf{Pa}(X_i) = \mathbf{z})$. These probabilities are estimated on disjoint sets of time steps to make the estimators independent of each other which we require while showing that the estimator is unbiased (Lemma G.1 in App. G). To estimate the above mentioned probabilities, we focus on the subsets

$$O_t^{x,z} = \{s \in O_t \mid X_i(s) = x, \mathbf{Pa}(X_i)(s) = \mathbf{z}\} \subseteq O_t$$

Let $C_t^x$ be the minimum value of $|O_t^{x,z}|$ (as $\mathbf{z}$ is varied). To use time steps in $E_t$ for estimating $\mathbb{P}(\mathbf{Pa}(X_i) = \mathbf{z})$, we partition this set into $C_t^x$ many parts[10], say $E_t = E_{t,1} \cup \dots \cup E_{t,C_t^x}$. For each part $E_{t,c}$, $c \in [C_t^x]$, we create an estimator of the probability $\mathbb{P}(\mathbf{Pa}(X_i) = \mathbf{z})$ as follows:

$$\widehat{p}_{t,c}^{\mathbf{z}} = \sum_{s \in E_{t,c}} \frac{\mathbb{1}\{\mathbf{Pa}(X_i)(s) = \mathbf{z}\}}{|E_{t,c}|}$$

Now we are ready to build an estimator using Equation 4. Let $s_1, \dots, s_{C_t^x}$ be any distinct elements[11] of set $O_t^{x,z}$. For each $c \in [C_t^x]$, we define a variable $Y_c^x$ as follows:

$$Y_c^x = \sum_{\mathbf{z}} \mathbb{1}\{Y(s_c) = 1\}\widehat{p}_{t,c}^{\mathbf{z}}$$

---

[10]Each part has at least $\lfloor |E_t|/C_t^x \rfloor$ elements. Choice of $C_t^x$ helps in bounding regret (Lemma G.2 in App. G).

[11]They exist since $|O_t^{x,z}| \geq C_t^x$.

Let $S_t^{i,x}$ be the set of timestamps $s \leq t$, when arm $a_{i,x}$ is pulled. Our final empirical estimator $\widehat{\mu}_{i,x}(t)$ of arm $a_{i,x}$ is:

$$\widehat{\mu}_{i,x}(t) = \frac{\sum_{s \in S_t^{i,x}} \mathbb{1}\{Y(s) = 1\} + \sum_{c \in [C_t^x]} Y_c^x}{N_t^{i,x} + C_t^x} \qquad (5)$$

It is easy to see that $\mathbb{E}[\widehat{\mu}_0(t)] = \mu_0$, and in Lemma G.1 (App. G) using backdoor criterion (Sec. 3.3.1 in Pearl [2009]) we show that $\mathbb{E}[\widehat{\mu}_{i,x}(t)] = \mu_{i,x}$ for every $i, x$. Finally, CRM-ALG uses Equations 3 and 5 to compute the UCB estimates $\bar{\mu}_{i,x}(t)$ and $\bar{\mu}_0(t)$ of arms $a_{i,x}$ and arm $a_0$ respectively

$$\bar{\mu}_{i,x}(t) = \widehat{\mu}_{i,x}(t) + \sqrt{\frac{2\ln t}{N_t^{i,x} + C_t^x}} \qquad (6)$$

$$\bar{\mu}_0(t) = \widehat{\mu}_0(t) + \sqrt{\frac{2\ln t}{N_t^0}} \qquad (7)$$

We bound the expected cumulative regret of CRM-ALG in Thm. 5.1, where $a^* = \arg\text{-}\max_{a \in A} \mu_a$ and, for $a \in \mathcal{A}$, $\Delta_a = \mu_{a*} - \mu_a$, $p_{\mathbf{z}}^{i,x} = \mathbb{P}(X_i = x, \mathbf{Pa}(X_i) = \mathbf{z})$, $p_{i,x} = \min_{\mathbf{z}} p_{\mathbf{z}}^{i,x}$. Additionally, $\eta_T^{i,x}$ denotes the probability that the empirical estimate of $p_{i,x}$ at time $T$ is large (See Observation G.6 in App. G) and is defined as $\eta_T^{i,x} = \max\{0, (1 - Z_i T^{-\frac{p_{i,x}^2}{4}})\}$, where $Z_i$ is the size of the domain of $\mathbf{Pa}(X_i)$.

**Theorem 5.1**
*If $a^* = a_0$, then at the end of $T$ rounds the expected cumulative regret is $O(1)$. Otherwise, the expected cumulative regret is of the order $\frac{58\ln T}{\Delta_0} + \Delta_0 + \sum_{\Delta_{i,x}>0} \Delta_{i,x} \max\Big\{0, 1 + 8\ln T\Big(\frac{1}{\Delta_{i,x}^2} - \frac{p_{i,x} \cdot \eta_T^{i,x}}{36\Delta_0^2}\Big)\Big\} + \sum_{\Delta_a>0} \Delta_a \frac{\pi^2}{3}$.*

The proof of Thm. 5.1 is given in App. G. Notice that the regret guarantee in Thm. 5.1 is an instance dependent constant if $a_0$ is optimal and otherwise slightly better than the UCB family of algorithms. Also, it is easy to construct examples of CBNs where the observational arm is optimal, for example see Experiment 2 in Sec. 6.

**No unobserved variable assumption:** As mentioned, in CRM-ALG we work in the fully observable setting unlike SRM-ALG (Sec. 3). A natural question is whether Algorithm C.1 (App. C) can also be used in CRM-ALG in the presence of unobserved confounders. We believe there is no straight-forward way to accomplish this due to a rather technical reason. Our estimator (Equation 5) cleverly interprets observational samples as $C_t^x$ many interventional samples and can be shown to be unbiased (Lemma G.1, App. G). The technique in Algorithm C.1 does not enable the same interpretation of observational samples, and hence cannot be easily used to create an estimator with similar properties.

# 6 EXPERIMENTS

In this section, we validate our results empirically. In Experiment 1, we compare the simple regret of our proposed algorithm SRM-ALG to two baseline MAB algorithms: uniform exploration (UE) and successive rejects (SR) [Audibert et al., 2010]. In Experiment 2, we compare SRM-ALG with a simple regret minimization algorithm for CB (referred to as PROP-INF or Propagating Inference from here onwards), given in Algorithm 3 in Yabe et al. [2018]. While implementing PROP-INF as described in Yabe et al. [2018], we faced multiple issues that we had to resolve. Details are provided in App. H. In Experiment 3, we compare the simple regret of SRM-ALG with baselines UE, SR as $m$ increases. In Experiment 4, we compare expected cumulative regret of CRM-ALG and UCB [Auer et al., 2002] when observational arm is the best arm ($a^* = a_0$) validating first part of Thm. 5.1. In Experiment 5, we compare expected cumulative regret of CRM-ALG and UCB on random CBNs with $a^* \neq a_0$ validating second part of Thm. 5.1.

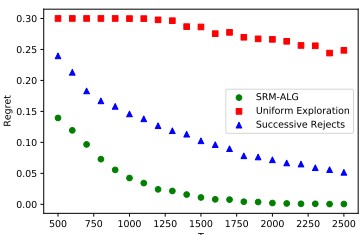

(a) SRM-ALG vs.UE ,SR

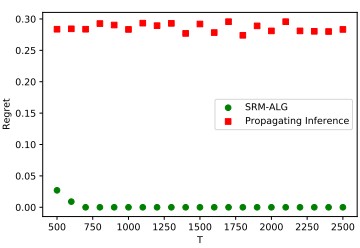

(b) SRM-ALG vs.PROP-INF

Figure 2: Simple Regret vs. $T$

**Experiment 1 (Simple Regret vs. $T$, SRM-ALG vs. UE, SR):** This experiment compares the expected simple regret of SRM-ALG with UE and SR as $T$ increases. We run the algorithms on 50 CBNs, where every constructed CBN $C$, has 100 intervenable nodes with $m(C) = 9$. The CBNs are constructed as follows: a) randomly generate 50 DAGs on 101 nodes $X_1, \ldots, X_{100}, Y$ such that $X_1 \prec \ldots \prec X_{100} \prec Y$ is a topological order in each such DAG, b) $\mathbf{Pa}(X_i)$ contains $\leq 2$ nodes chosen uniformly at random from $X_1, \ldots, X_{i-1}$, and $\mathbf{Pa}(Y)$ equals the set of all $X_i$s, c) $\mathbb{P}(X_i \mid \mathbf{Pa}(X_i)) = 0.5$ for $i \in$ [91] and $\mathbb{P}(X_i|\mathbf{Pa}(X_i)) = 1/18$ for $i \in [92, 100]$, d)

uniformly at random choose a $j \in \{92, \ldots, 100\}$ and set $P(Y|X_1, \ldots, X_j = 1, \ldots, X_{100}) = 0.5 + \epsilon$ and $P(Y|X_1, \ldots, X_j = 0, \ldots, X_{100}) = 0.5 - \epsilon'$ where $\epsilon = 0.3$ and $\epsilon' = q\epsilon/(1-q)$ for $q = 1/18$. Our choice of the conditional distributions in (c) ensures $m(C) = 9$ for every generated CBN $C$. Our strategy to generate CBNs is a generalization of the one used in Lattimore et al. [2016]. For each of the 50 CBN, we run SRM-ALG, MAB, SR for multiple values of the time horizon $T$ in $[500, 2500]$ and average the regret over 100 independent runs. We average the regret over the 50 CBNs and plot mean regret vs. $T$ in Fig. 2a. Since $m \ll N$, we see that, SRM-ALG has a much lower regret compared to UE,SR in accordance with Thm. 3.1.

**Experiment 2 (Simple Regret vs. $T$, SRM-ALG vs. PROP-INF):** This experiment compares the expected simple regret of SRM-ALG with CB as $T$ increases. We run the algorithms on 50 CBNs such that for every constructed CBN $C$, it has 10 intervenable nodes and $m(C) = 5$. The CBNs are constructed as follows: a) randomly generate 50 DAGs on 11 nodes $X_1, \ldots, X_{10}$ and $Y$, and let $X_1 \prec \ldots \prec X_{10} \prec Y$ be the topological order in each such DAG, b) $\mathbf{Pa}(X_i)$ contains at most 1 node chosen uniformly at random from $X_1, \ldots, X_{i-1}$, and $\mathbf{Pa}(Y) = \{X_6, \ldots, X_10\}$, c) $\mathbb{P}(X_i \mid \mathbf{Pa}(X_i)) = 0.5$ for $i \in [5]$ and $\mathbb{P}(X_i|\mathbf{Pa}(X_i)) = 1/10$ for $i \in [6, 10]$, d) uniformly at random choose a $X_j$ from $\mathbf{Pa}(Y)$ and set the CPD of $Y$ as $\mathbb{P}(Y|\ldots, X_j = 1, \ldots) = 0.5 + \epsilon$ and $\mathbb{P}(Y|\ldots, X_j = 0, \ldots) = 0.5 - \epsilon'$ where $\epsilon = 0.3$ and $\epsilon' = q\epsilon/(1-q)$ for $q = 1/10$. The choice of the conditional probability distributions (CPDs) in (c) ensures $m(C) = 5$ for every CBN $C$ that is generated. Our strategy to generate CBNs is a generalization of of the one used in Lattimore et al. [2016] to define parallel bandit instances with a fixed $m$. For each of the 50 random CBN, we run SRM-ALG and CB for multiple values of the time horizon $T$ in $[500, 2500]$ and average the regret over 30 independent runs. We calculate the mean regret over the 50 random CBNs and plot mean regret vs. $T$ in Fig. 2b. As seen, SRM-ALG has a much lower regret compared to PROP-INF which incurs $\tilde{O}(\sqrt{N/T})$ regret in comparison to SRM-ALG's regret of $\tilde{O}(\sqrt{m/T})$ (Theorem 3.1).

**Experiment 3 (Simple Regret vs. $m$):** This Exp. compares the expected simple regret of SRM-ALG with UE and SR for CBNs with different values of function $m$ from the set $M = \{10 + 2k : k \in [20]\}$. For this experiment, we fix the time horizon to $T = 1600$. We randomly generate 35 DAGs on $N + 1$ nodes $X_1, \ldots, X_N$ and $Y$. For each generated DAG $\mathcal{G}$ and $m \in M$, we use the same process as Exp. 1 to set the CPDs of $\mathcal{G}$. For each of the 35 random CBNs thus obtained, we run SRM-ALG, MAB, SR for time horizon $T$ and average the regret over 100 independent runs. We repeat this Exp. for $N = 100$ and $N = 200$. For $N = 100$, we plot the mean regret over all the 35 random CBNs vs. $m$ in Fig. 3a. The same plot for $N = 200$ is provided in Fig. 3b. Our plots validate the $\sqrt{m}$ dependence of regret (for fixed

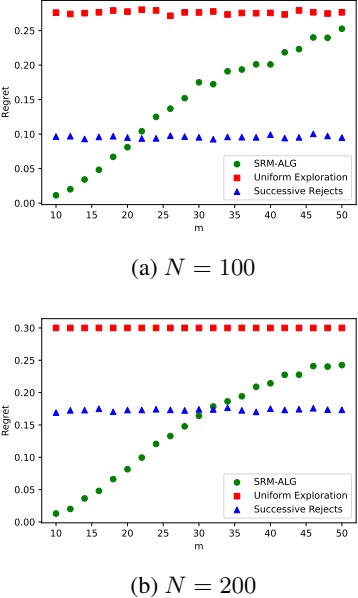

(a) $N = 100$

(b) $N = 200$

Figure 3: Simple Regret vs. $m$

$T$) in the case of SRM-ALG. We see that as $N$ increases (with $m$ fixed), regret of SRM-ALG is constant (as shown in Theorem 3.1), whereas regret of MAB and SR increases (as indicated by their regret guarantees). Thus, for large $N$, SRM-ALG is strictly better, for a wide range of values of $m$.

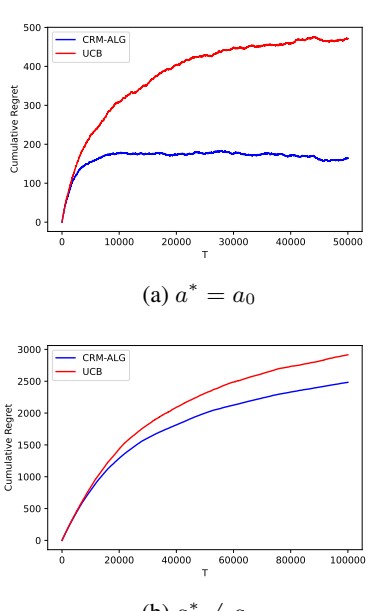

(a) $a^* = a_0$

(b) $a^* \neq a_0$

Figure 4: Cumulative Regret vs. $T$

**Experiment** 4 **(Cumulative Regret vs. T, $a^* = a_0$):** In this experiment, we compare cumulative regret of CRM-ALG with UCB for CBN on four nodes $X_1, X_2, X_3$, and $Y$. $X_1$ has no parents and is the only parent of $X_2, X_3$. Parents

of $Y$ are $X_2, X_3$. We choose CPDs: $\mathbb{P}(X_1 = 1) = 0.5$, $\mathbb{P}(X_2 = 1|X_1)$ and $\mathbb{P}(X_3 = 1|X_1)$ are equal to $0.75X_1 + 0.25(1 - X_1)$ and $P(Y = 1|X_2, X_3) = \mathbb{1}_{X_2 = X_3}$. For this instance, it is easy to see that $\mathbb{P}(Y = 1|do(X_2 = x)) = P(Y = 1|do(X_3 = x)) = 0.5$ for $x \in \{0, 1\}$ and $P(Y = 1|do()) = 5/8$, implying that observational arm is the best arm. We average the cumulative regrets of CRM-ALG and UCB over 30 independent runs. Fig. 4a demonstrates that cumulative regret of UCB increases and that of CRM-ALG becomes constant for large $T$ (as shown in Thm. 5.1).

**Experiment** 5 **(Cumulative Regret vs. T, $a^* \neq a_0$):** This experiment compares the cumulative regret of CRM-ALG with UCB as $T$ increases. The algorithms are run on 12 CBNs such that for every constructed CBN $\mathcal{C}$, it has 10 intervenable nodes. The CBNs are constructed as follows: a) randomly generate 12 DAGs on 11 nodes $X_1, \dots, X_{10}$ and $Y$, and let $X_1 \prec \dots \prec X_{10} \prec Y$ be the topological order in each such DAG, b) $\mathbf{Pa}(X_i)$ contains at most 1 node chosen uniformly at random from $X_1, \dots, X_{i-1}$, and $\mathbf{Pa}(Y)$ contains $X_i$ for all $i$, c) $\mathbb{P}(X_i \mid \mathbf{Pa}(X_i)) = 0.5$ for $i \in [10]$ and, d) uniformly at random choose a $X_j$ from $\mathbf{Pa}(Y)$ and set the CPD of $Y$ as $\mathbb{P}(Y \mid \dots, X_j = 1, \dots) = 0.5 + \epsilon$ and $\mathbb{P}(Y \mid \dots, X_j = 0, \dots) = 0.5 - \epsilon'$ where $\epsilon = 0.1$ and $\epsilon' = q\epsilon/(1 - q)$ for $q = 1/2$, that is an interventional arm is the best arm. We average the cumulative regrets of CRM-ALG and UCB over 30 independent runs. Fig. 4b demonstrates that cumulative regret of CRM-ALG gets better than that of UCB for large $T$ (as shown in Thm. 5.1).

# 7 CONCLUSION

We proposed two algorithms SRM-ALG and CRM-ALG that take as input causal graphs, pull observational/interventional arms, and minimize simple and cumulative regret respectively. While SRM-ALG works over SMCGs and can handle unobserved variables, CRM-ALG works in the fully observable setting. We theoretically and empirically show that our algorithms are better than standard MAB algorithms that do not take causal side-information into account. Further, we show that SRM-ALG is almost optimal for causal graphs having an $n$-ary tree structure. In the fully observable setting, our algorithms do not put any restrictions on the graph structure and subsume previous results which imposed strong structural restrictions. We plan to explore cumulative regret minimization in the presence of UCs in a future work. Another interesting direction is to identify graphs where better simple regret guarantee than SRM-ALG can be attained. Finally, obtaining regret guarantees when interventions are non-atomic will be a nice extension to our work.

# 8 AUTHOR CONTRIBUTION

Aurghya Maiti is the lead author of the paper. Vineet Nair and Gaurav Sinha contributed equally.

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
