# OpenReview forum: "A Causal Bandit Approach to Learning Good Atomic Interventions in Presence of Unobserved Confounders"
_auai.org/UAI/2022/Conference — UAI 2022 Poster_

### Official Review · Reviewer_bnuR · 2022-04-01

**Q2(1) Originality/Novelty:** 2
**Q2(2) Significance/Impact:** 2
**Q2(3) Correctness/Technical Quality:** 3
**Q2(6) Clarity Of Writing:** 3
**Q6 Overall Score:** 4
**Q8 Confidence In Your Score:** 4

**Q1 Summary And Contributions:**

The manuscript introduces two multi-armed bandit algorithms that exploit causal structure to minimize simple and cumulative regret. Theoretical properties of these methods are studied, including a near optimality result under certain constraints.

**Q2 Assessment Of The Paper:**

More detailed information regarding each of these aspects is given below:

**Q2(4) Quality Of Experiments (Optional):**

2: Fair: The experimental evaluation is weak: important baselines are missing, or the results do not adequately support the main claims.

**Q2(5) Reproducibility:**

3: Good: Key resources (e.g., proofs, code, data) are available and key details (e.g., proofs, experimental setup) are sufficiently well-described for competent researchers to confidently reproduce the main results.

**Q3 Main Strengths:**

The discussion of causal approaches to bandit problems is well-motivated and the proposed algorithms are thoroughly described. Though I did not go through the proofs in close detail, the results appear sound.

**Q4 Main Weakness:**

It was not entirely clear to me how these algorithms go beyond existing work. This line in particular struck me as problematic: “To the best of our knowledge, this is the first work that analyses the regret of causal bandit algorithms when input causal graphs contains UCs [unobserved confounders].” Elias Bareinboim has been publishing on this topic for years, including in several papers cited in this manuscript’s bibliography (e.g., the 2015 NeurIPS paper “Bandits with unobserved confounders” and the 2018 NeurIPS paper “Structural causal bandits: Where to intervene?”). How, if at all, do SRM-ALG or CRM-ALG improve upon these methods?

The experiments section is brief and unsatisfying. As far as I can tell, it includes no other causal bandit benchmarks, even though several such methods exist and are discussed in the manuscript.

**Q5 Detailed Comments To The Authors:**

It is possible I have simply misunderstood something about the proposal here that distinguishes it from existing methods, but if so then this should be clearly stated in the manuscript. Further experiments could also help drive home the unique selling points of this method. The technical discussion appears well executed, though it is hard for me to evaluate this as I am not an expert on bandit algorithms. That said, I am aware of other work in this area that aims at similar goals, some of which is explicitly mentioned in the manuscript. The text could do a better job distinguishing the present proposal from those works, both through theory and experiments. As it stands, the claim that no other causal bandit method considers the case of unobserved confounding is simply false.

Minor comments:
-I would recommend adding a note to Sect. 2 (notation) on the use of Y(t) to denote response at round t, as this is often denoted with subscripts, i.e. Y_t. My first thought seeing Y(t) was potential outcomes notation, i.e. shorthand for Y | do(T = t), which obviously makes no sense in this context.

-The dash in “arg - max” is somewhat jarring. These should just be “argmax”.


**Q7 Justification For Your Score:**

Despite this paper's merits, I am uncertain how the algorithms proposed here go beyond existing work in this area, especially those of Bareinboim and colleagues. I would consider revising my score upward if the manuscript engaged with these issues more thoroughly, and expanded experiments to include more comprehensive benchmarks.

**Q9 Complying With Reviewing Instructions:**

1: Yes.

---

### Official Review · Reviewer_iGBD · 2022-04-09

**Q2(1) Originality/Novelty:** 2
**Q2(2) Significance/Impact:** 2
**Q2(3) Correctness/Technical Quality:** 3
**Q2(6) Clarity Of Writing:** 3
**Q6 Overall Score:** 6
**Q8 Confidence In Your Score:** 4

**Q1 Summary And Contributions:**

This paper considers bandit algorithms where the underlying reward mechanism can be described as a CBN. In particular, the agent can only access to the model-induced causal graph. The authors proposed algorithms for i) identifying a best arm and ii) minimizing cumulative regret by intervening on a single variable or none (observational sampling). By taking advantage of causal effect identifiability, the proposed algorithms perform better than other algorithms without side causal information.


**Q2 Assessment Of The Paper:**

More detailed information regarding each of these aspects is given below:

**Q2(4) Quality Of Experiments (Optional):**

3: Good: The experimental evaluation is adequate, and the results convincingly support the main claims.

**Q2(5) Reproducibility:**

3: Good: Key resources (e.g., proofs, code, data) are available and key details (e.g., proofs, experimental setup) are sufficiently well-described for competent researchers to confidently reproduce the main results.

**Q3 Main Strengths:**

The two algorithms with better performance, rigorous regret analysis, and corroborating empirical results.  The result seems useful in understanding a very specific type of causal graph.

The paper's topic is highly relevant.

**Q4 Main Weakness:**

The assumption that every intervenable variable having no bidirected path to all of its children is very restrictive in that you can always obtain every arm’s reward distribution P(y|do(x)) from P(v). In fact, the assumption is even stronger. You can identify the whole distribution P(v\x|do(x)). The wording that “with unobserved confounders” doesn’t seem sufficient.

The significance of the result is a bit limited due to the type of graph considered.

**Q5 Detailed Comments To The Authors:**

Correctness: The paper appears to be correct where causal inference is used. I skipped regret analysis part which is not my expertise.
Literature Review: Some of the recent work on causal bandits and differences between this work and existing work are well discussed.
Clarity: The paper is generally clearly written.   I have reviewed this paper before and almost all of my concerns are now addressed.

In Section 5, every variable is now observable where backdoor criterion can be used. I wonder whether the backdoor criterion with parents as admissible set is the most desirable choice. There may be a different adjustment set that leads to lower variance in causal effect estimation than the parents. By the way, you don’t have to assume that every variable is observable, but you can just say every P(y|do(X_i)) is identified by adjusting for its parents, which is less restricting.

Again, if I am not mistaken, atomic intervention represents hard intervention (do) where its opposite is soft intervention, conditional intervention, etc. Singleton intervention would be a reasonable choice.

Missing period (footnote 2, 3, 6, 9)

(Disclaimer: I've reviewed the previous version of this paper.)

**Q7 Justification For Your Score:**

While the paper doesn't address fully general graphs where some of the causal effects are not identifiable, the result here seems a good starting point to further explore the regret analysis of causal bandits for general graphs.

**Q9 Complying With Reviewing Instructions:**

1: Yes.

---

### Official Review · Reviewer_xPSF · 2022-04-11

**Q2(1) Originality/Novelty:** 3
**Q2(2) Significance/Impact:** 3
**Q2(3) Correctness/Technical Quality:** 3
**Q2(6) Clarity Of Writing:** 3
**Q6 Overall Score:** 7
**Q8 Confidence In Your Score:** 4

**Q1 Summary And Contributions:**

The paper presents an extension of previous work for causal bandits when there are unobserved confounders. It provides algorithms for simple regret minimization with confounders and for multiple regret minimization without confounders, and also some theoretical guarantees. Experimental results compare the proposed algorithms with some baselines showing superior performance.

**Q10 Ethical Concerns (Optional):**

No ethical concerns.

**Q2 Assessment Of The Paper:**

More detailed information regarding each of these aspects is given below:

**Q2(4) Quality Of Experiments (Optional):**

3: Good: The experimental evaluation is adequate, and the results convincingly support the main claims.

**Q2(5) Reproducibility:**

3: Good: Key resources (e.g., proofs, code, data) are available and key details (e.g., proofs, experimental setup) are sufficiently well-described for competent researchers to confidently reproduce the main results.

**Q3 Main Strengths:**

The paper advances in the use of causal models for bandits problem, considering unobserved confounders not considered in previous work.
It is well presented, including an extensive additional material with demostration and additional experiments.

**Q4 Main Weakness:**

One issue is why not considered confounders for the case of cumulative regret, the explanation given is not convincing.

**Q5 Detailed Comments To The Authors:**

Clarify /extend the explanation for not considering confounders in the case of accumalative regret.
Discuss the condition of m(C) << N, what does this imply? It is common in practice?

**Q7 Justification For Your Score:**

The paper contributes in the use of causal models for bandit problems, extending previous work for the case of unobserved confounders; this could have impact in other applications of causal models such as reinforcement learning.

**Q9 Complying With Reviewing Instructions:**

1: Yes.

---

### Official Review · Reviewer_Fo8q · 2022-04-13

**Q2(1) Originality/Novelty:** 3
**Q2(2) Significance/Impact:** 3
**Q2(3) Correctness/Technical Quality:** 3
**Q2(6) Clarity Of Writing:** 3
**Q6 Overall Score:** 7
**Q8 Confidence In Your Score:** 3

**Q1 Summary And Contributions:**

The submission studies the problem of determining the highest-value atomic intervention given coarse-grained causal background knowledge allowing for a degree of unobserved confounding. The authors propose a causal bandit approach and study its regret profile, showing theoretical and experimentally that it compares very favorably with the state of the art.

**Q2 Assessment Of The Paper:**

More detailed information regarding each of these aspects is given below:

**Q2(4) Quality Of Experiments (Optional):**

3: Good: The experimental evaluation is adequate, and the results convincingly support the main claims.

**Q2(5) Reproducibility:**

3: Good: Key resources (e.g., proofs, code, data) are available and key details (e.g., proofs, experimental setup) are sufficiently well-described for competent researchers to confidently reproduce the main results.

**Q3 Main Strengths:**

- Important, interesting problem and innovative solution concept.
- Good theoretical and experimental justification.


**Q4 Main Weakness:**

- Some unclarity on technical and theoretical matters.

**Q5 Detailed Comments To The Authors:**

The submission studies the problem of determining the highest-value atomic intervention given coarse-grained causal background knowledge allowing for a degree of unobserved confounding. The authors propose a causal bandit approach and study its regret profile, showing theoretically and experimentally that it compares favorably with the state of the art. A rigorous and innovative approach to a difficult and important problem. I have two questions that I would like addressed:


1. I am not clear to what degree we are really allowing for unobserved confounding here. In the third graf of Section 1, the authors write that they are studying causal graphs with unobserved variables that are parents of *at least* two observable variables. But in Section 1.2, we seem to assume that every unobserved variable has *exactly* two children. Have we really not lost any generality here?

More importantly, the authors assume that the effect of $do(X=x)$ can be consistently estimated from the observational data alone. According to the result of Tian and Pearl appealed to by the authors, this effect is the case if there is no bi-directed path connecting $X$ to any of its children. But $X$ and $Y$ have no bi-directed path between them iff they have no unobserved parent. So haven’t we assumed away the relevant kind of confounding by assuming identifiability? Haven’t we just assumed that the intervention variable and the target variable have no unobserved common cause? What am I missing?

2. I do not quite see whether the authors are modeling any observation-intervention trade-off. The algorithm always does the same number of pulls of the observational and interventional arms, so I suppose this tradeoff is unmodeled. Can the authors discuss how the approach might be extended to handle this in a subtler way?

3. Am I right that the experiments compare the proposed algorithm only with algos that use no causal information at all? Is this really the relevant comparison? Shouldn’t they be compared with other causal bandit algos (for example those that don’t account for unobserved confounders)?

Minor Issues:

- I don't think the example in Figure 1 is very good, since it is very implausible to implement a surgical intervention in this setting. There is no way to break the edge from social-economic variables to working from home. Care workers etc. cannot work from home.

- I could use a reminder in the text on the difference between big-O vs. big-Omega, as written the comparisons in e.g. the second graf on section 1.1 don’t look apple-to-apple.


**Q7 Justification For Your Score:**

Innovative approach to an important problem. Although theoretical issues remain, I think it should be accepted given some issues are cleared up.

**Q9 Complying With Reviewing Instructions:**

1: Yes.

---

### Decision · Program_Chairs · 2022-05-15

**Decision:**

Accept (Poster)

**Comment:**

Meta Review: Thank you for your submission to UAI, and the authors' response to reviewers' concerns.

This paper demonstrates a causal bandit method for determining the best atomic intervention given knowledge of the causal graph governing a system.  The paper presents algorithms for simple regret minimization with confounders, with both theoretical analysis and experimental validation.  Reviewers appreciated the work's innovativeness and potential broader applications, such as in reinforcement learning.